# Cyclostratigraphy of the Middle to Upper Ordovician successions of the Armorican Massif (western France) using portable X-ray fluorescence

Matthias Sinnesael[1,2,3], Alfredo Loi[4], Marie-Pierre Dabard[5*], Thijs R.A. Vandenbroucke[2], Philippe Claeys[1]

[1] Analytical, Environmental and Geo-Chemistry, Vrije Universiteit Brussel, Pleinlaan 2, 1050 Brussels, Belgium
[2] Department of Geology, Ghent University, Krijgslaan 281/S9, 9000 Ghent, Belgium
[3] IMCCE, CNRS, Observatoire de Paris, PSL University, Sorbonne Université, 77 Avenue Denfert-Rochereau, 75014, Paris, France
[4] University of Cagliari Department of Chemical and Geological Sciences, Cittadella Universitaria, Blocco A – 09042, Monserrato, Italy
[5] Géosciences UMR6118 CNRS/Université Rennes1, Campus de Beaulieu, 35042 Rennes, Cédex, France
[*] Deceased.

*Correspondence to*: Matthias Sinnesael (matthias.sinnesael@obspm.fr)

**Abstract.** To expand traditional cyclostratigraphic numerical methods beyond their common technical limitations and apply them to truly deep-time archives we need to reflect on the development of new approaches to sedimentary archives that traditionally are not targeted for cyclostratigraphic analysis, but that frequently occur in the impoverished deep-time record. Siliciclastic storm-dominated shelf environments are a good example of such records. Our case study focusses on the Middle to Upper Ordovician siliciclastic successions of the Armorican Massif (western France), which are well-studied in terms of sedimentology and sequence stratigraphy. In addition, these sections are protected geological heritage due to the extraordinary quality of the outcrops. We therefore tested the performance of non-destructive high-resolution (cm-scale) portable X-ray fluorescence and natural gamma-ray analyses on outcrop to obtain major and trace element compositions. Despite the challenging outcrop conditions in the tidal beach zone, our geochemical analyses provide useful information regarding general lithology and several specific sedimentary features such as the detection of paleoplacers, or the discrimination between different types of diagenetic concretions such as nodules. Secondly, these new high-resolution data are used to experiment with the application of commonly used numerical cyclostratigraphic techniques on this siliciclastic storm-dominated shelf environment, a non-traditional sedimentological setting for cyclostratigraphic analysis. In the lithological relatively homogenous parts of the section spectral power analyses and bandpass filtering hint towards a potential astronomical imprint of some sedimentary cycles, but this needs further confirmation in the absence of more robust independent age constraints.

# 1 Introduction

Astronomical climate forcing is a major driver of natural climate change and its sedimentological expressions can often be identified in the stratigraphic record (Hinnov, 2018; Meyers, 2019; Sinnesael et al. 2019; Laskar, 2020). Studying past astronomical imprints is informative of past climate dynamics and sedimentological processes. Moreover, due to the almost semi-periodic nature of the astronomical cycles of eccentricity, precession and obliquity, the identification of theses cycles (i.e. cyclostratigraphy) also provides highly-resolved temporal constraints to sedimentary sequences (i.e. astrochronology). Cyclostratigraphic analyses are performed on a wide range of sedimentological settings (Meyers, 2019; Sinnesael et al., 2019). Notwithstanding, most of the currently used techniques for numerical analysis imply stringent underlying assumptions regarding the nature of the record; i.e. complete continuous records without major changes in accumulation rates or depository environments (Weedon, 2003; Hilgen et al., 2015). Typically, open marine pelagic deposits are considered as the ideal stratigraphic record for this type of work (e.g. Westerhold et al., 2020). However, the further back in geological time, the lesser these records are preserved, and they are virtually absent in the pre-Cretaceous record. Therefore, there is a need to explore new ways of identifying astronomical cycles in less traditional archives using a broader range of analysis techniques (Noorbergen et al., 2018; Lantink et al., 2019; Montanez, 2021).

This study compares numerical time-series analysis (e.g. Weedon, 2003) and an alternative approach based on relative sea-level interpretations (e.g. Cataneanu et al., 2009), applied to a siliciclastic storm-dominated shelf environment, a setting not traditionally targeted by cyclostratigraphic studies. The Ordovician sections of the Crozon Pensisula (France; Fig. 1) have been studied in detail for cyclic sedimentary expressions by interpreting stacked stratigraphic sequences (Fig. 2; Dabard et al., 2015 and references therein). Previous work has resulted in detailed sea-level change reconstructions based on sedimentological interpretations and low-resolution (m-scale) natural gamma-ray data (Dabard et al., 2007; 2015). Backstripping was applied to identify subsidence and several orders of sea-level change (Fig. 2). The 3rd to 5th order eustatic sea-level changes are hypothesized to correspond to various frequencies related to astronomical forcing. Here, we expand the existing data set with new medium resolution (dm-scale) natural gamma ray (NGR) and high-resolution (cm-scale) portable X-ray fluorescence (pXRF) data for two selected stratigraphical intervals with contrasting lithofacies. The high-resolution pXRF data can be used for detailed time-series analysis and provides additional geochemical insights on specific sedimentological features like nodules and paleoplacer beds (Loi and Dabard, 2002; Dabard and Loi, 2012; Pistis et al., 2008; 2016; 2018).

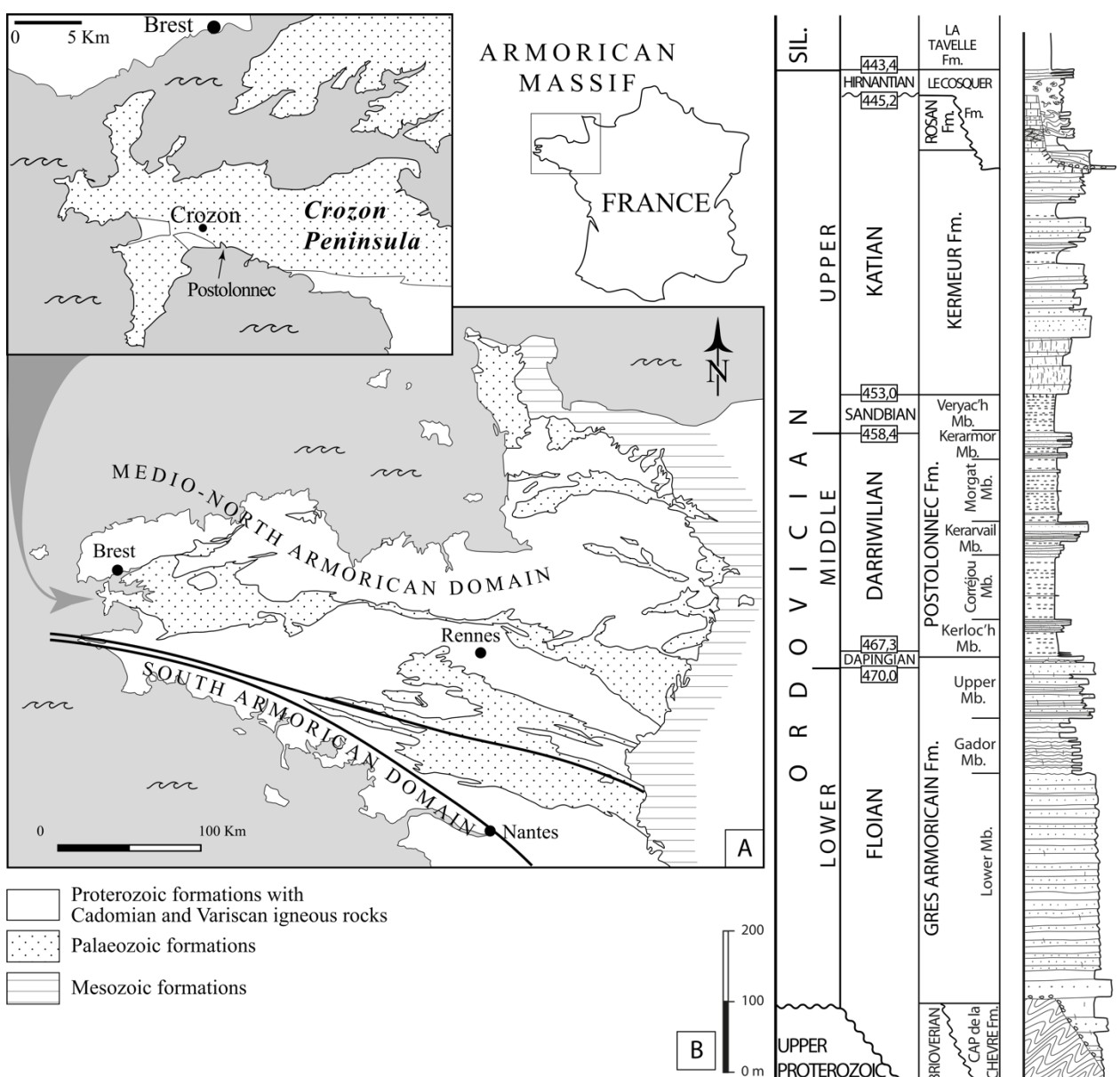

**Figure 1: (A) Location of the Postolonnec section on the Crozon Peninsula of the Armorican Massif. (B) Lithostratigraphic column of the Ordovician succession on the Crozon Peninsula. Modified from Dabard et al. (2015).**

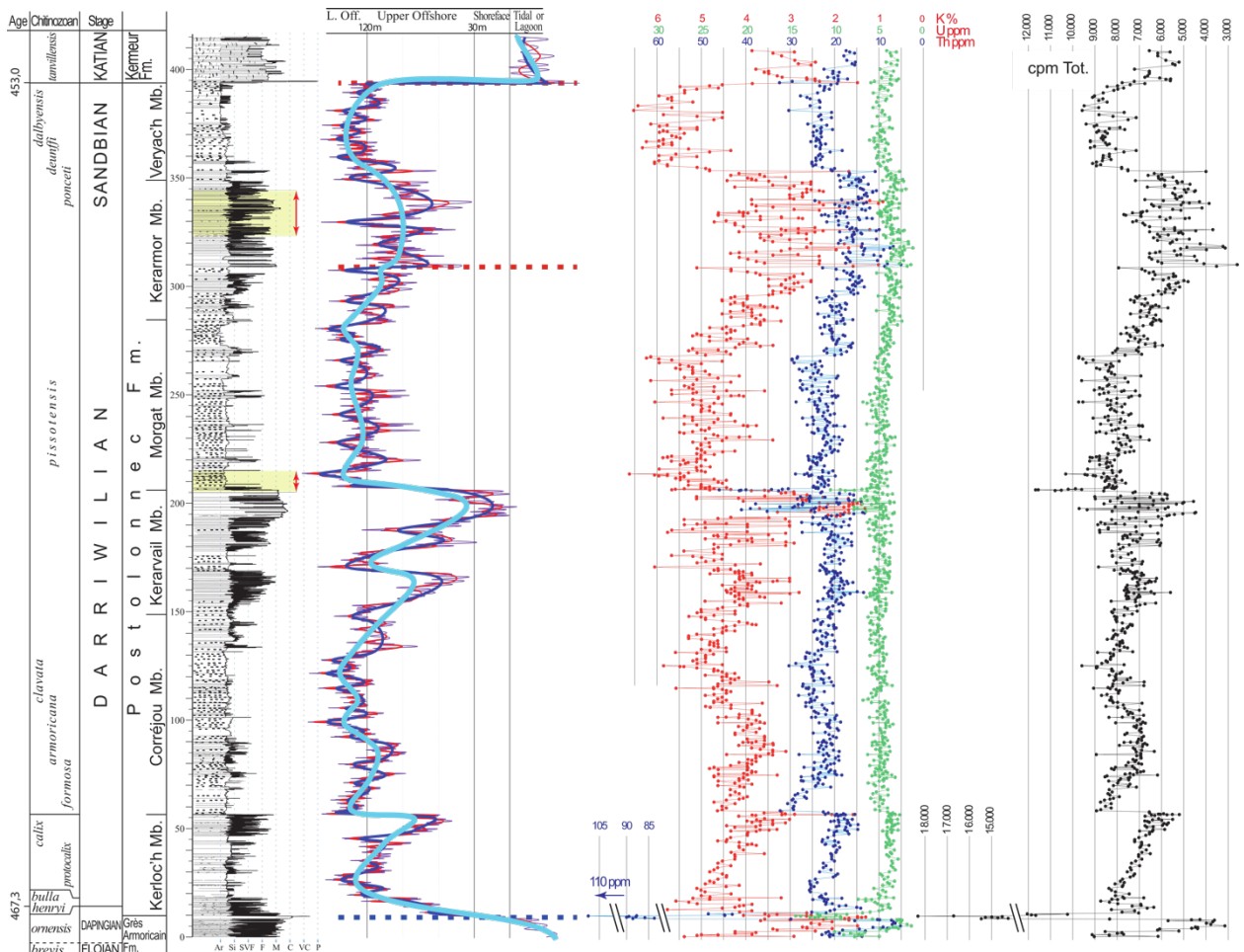

**Figure 2: Stratigraphy of the Postolonnec Formation including chitinozoan biostratigraphy, lithofacies interpretation, sequence stratigraphy (very high frequency (purple and red curves), high frequency or fourth-order (blue curve) and third-order (pale blue curve) sequences, transgressive ravinement surface (blue dotted line), sharp-based surfaces (red dotted lines)) and gamma-ray spectral logs (cpm = count per minute). Orange arrows and shading indicate sections studied in high resolution in this study in the Morgat and Kerarmor Members. Modified from Dabard et al. (2015).**

## 2 Geological setting and biostratigraphy

The Middle to Upper Ordovician sections of the Crozon Peninsula area, Armorican Massif (Paris et al., 1999; Vidal et al., 2011a), western France (Fig. 1A), were deposited on the continental terrigenous platform of Gondwana, located at high paleolatitudes (~60°S). This study focuses on the Postolonnec Formation (Fm.) (Fig. 1B) as it is outcropping at the coastal cliffs at Postolonnec (48°14'18.50'' N, 4°27'59.70'' W; see Fig. 3 in Dabard et al., 2015). The section is characterized by siliciclastic facies ranging from medium- to coarse-grained sandstone beds with hummocky cross stratification (HCS) to silty mudstones containing siltstone laminae and lenticular coquina beds (Fig. 3, Dabard et al., 2015). Sedimentary features like

HCS and lenticular coquina beds reflect storm wave action, and overall these sediments were deposited in storm-dominated shelf environments.

The biostratigraphic calibration is based on a standard Gondwanan chitinozoan biozonation (Fig. 2; Paris, 1981; 1990; Dabard et al., 2007; Dabard et al., 2015). *Desmochitina ornensis* supports a Dapingian position for the top of the Grès

Armoricain Formation and the base of the Postolonnec Fm. while the succeeding *Belonechitina henryi* Biozone of the lower Darriwilian is found several meters above the base of the Postolonnec Fm. Other chitinozoan biozones found in the Crozon area supporting a lower Darriwilian position are *Desmochitina bulla, Cyathochitina protocalix*, and *Siphonochitina formosa*. Specimens of *Linochitina pissotensis* are recorded in the Morgat Member (Mb.) and indicate the upper Darriwilian Stage. The base of the Sandbian is correlated with the *Lagenochitina ponceti* Biozone and the Veryac'h Member, which also yields

two younger Sandbian Biozones: the *Lagenochitina deunffi* and the *Lagenochitina dalbyensis* Biozones. Overall, the Postolonnec Fm. spans roughly fourteen million years starting close to the start of the Darriwilian Stage (~467 Ma) and ending close to the end of Sandbian Stage (~453 Ma). No radio-isotopic dating, chemostratigraphic or any other further constraints are currently available.

## 3 Methods

The coastal sections of Postolonnec are protected as valuable geological and natural heritage. Therefore, we investigated the potential use of non-destructive NGR and pXRF measurements on outcrop for stratigraphic and geochemical characterizations. Next to common challenges of using pXRF on outcrop (surface weathering and need for flat sample surface), a specific challenge for this beach section is that it is subject to large tidal ranges, which makes the outcrops wet for most of the day, and sometimes covered by organisms with carbonate shells (e.g. barnacles). Measurements were always

performed during dry weather, after low tide and on fresh uncovered outcrop surfaces. The potential added value of the pXRF compared to the handheld NGR is that pXRF gives a wider range of elemental analyses and has a finer spatial resolution regarding spot size (~1 cm vs. ~10 cm resolution). Moreover, the unshielded NGR measurement device integrates the natural radioactive field beyond its actual spot size (sphere with radius of several dm), which consequently, is influenced by the geometry of the outcrop at the point of measurement. Elemental variations measured by both instruments (e.g.

potassium) can also be compared to test the robustness of the signals. In contrast to the NGR, the pXRF cm-resolution allows us to measure fine-scale features such as individual nodules or paleoplacers (Loi and Dabard, 2002; Dabard and Loi, 2012; Pistis et al., 2016; 2018). Conversely, one has to keep in mind that because of this fine spot size the result of the measurement is very sensitive to the actual spot of analysis. For example, in a coarse-grained heterogenous sandstone one could imagine hitting a different mineralogy for different measurements within the same bed. The comparison of the

geochemical composition of the nodules and their immediately surrounding matrix, for example, could be useful to gain insight in their genesis. For our study, we compiled the acquired pXRF, NGR and lithology data, evaluated how they

compared to each other, and tested if the pXRF indeed reflects lithological and geochemical changes in these challenging measurement conditions.

Two field surveys were carried out analyzing two different stratigraphic intervals with contrasting facies in cm-resolution. The stratigraphically lower ~10-m-thick interval spans the transition from the shoreface facies Kerarvail Member (Mb.) into the deeper marine Morgat Member that is dominated by a deeper-marine clay-silt facies (Figs. 2-3). The stratigraphically higher ~14-m-thick interval in the Kerarmor Mb. is characterized by larger and more frequent variations in lithology, ranging from mudstones to medium-sized sandstones (Figs. 2-3). Both sections were logged at a 1:20 scale and later transformed into

a 1-cm resolution detailed sedimentary log (presented in simplified forms in the lithology columns of Fig. 4). The approach used in this work for the depositional sequence analyses and sea level reconstructions are the same as those used in Dabard et al. (2015) with the only difference being greater detail and a higher density of NGR measurements in both sandy and clayey homolithic facies in this study. Successively, we measured the sections in high-resolution with handheld NGR (10-20 cm resolution) and pXRF (1-10 cm resolution) (Fig. 4). Dabard et al. (2015) measured NGR for the whole Postolonnec Fm.

at a varying 10-100 cm resolution according to the thickness and homogeneity of the facies. Table 1 summarizes all available data. All data and detailed logs are provided in Supplementary Materials.

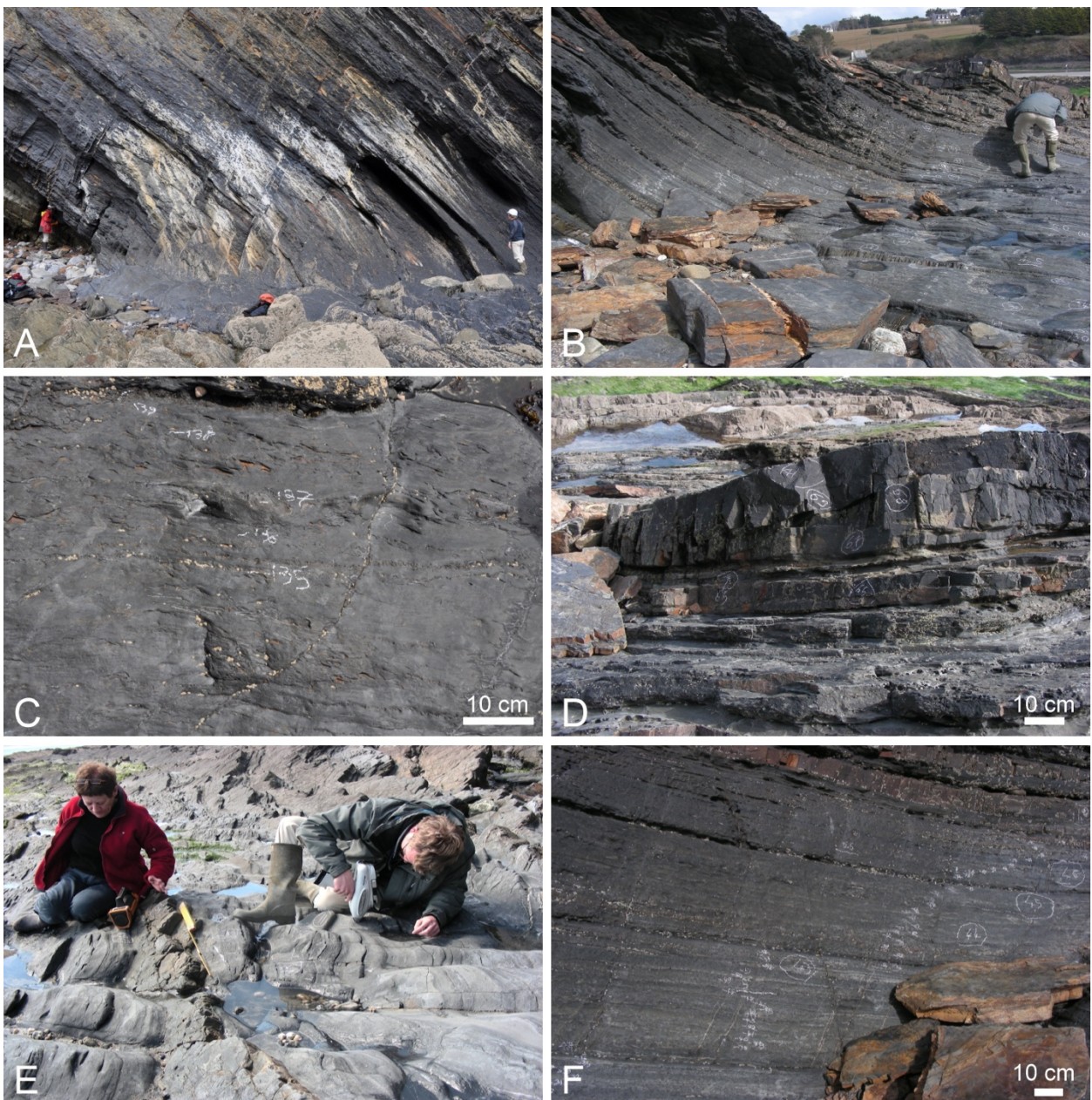

**Figure 3: Postolonnec Formation outcrop images. (A) Overview basal Morgat Member. (B) Overview of the upper half of the detailed logged section in the Kerarmor Member. (C) Detail of the condensed lithofacies of the Morgat Mb. with dark nodules and shell beds. White chalk numbers indicate locations of pXRF analyses. (D) Detail of Kerarmor sandstones with white chalk circles indicating locations of NGR analyses. (E) Modus operandi of the use of pXRF and NGR on the outcrop surface. (F) Detail of Fig. 3B illustrating the high-resolution measurement strategy.**

 **Table 1: Overview of available numerical data for the Postolonnec Formation. Stratigraphic intervals and members follow Dabard et al. (2015). NGR = Natural Gamma-Ray, pXRF = portable X-Ray Fluorescence.**

| Stratigraphic Interval (m) | Member | Data type | Resolution (cm) | points (N) | Source |
|---|---|---|---|---|---|
| 0.0-409.0 | Kerloc'h - Veryac'h | NGR | 10 to 100 | 883 | Dabard et al. (2015) |
| 204.5-214.5 | Morgat | Lithology | ~1 | 991 | This study (09/2017) |
| 204.5-214.5 | Morgat | NGR | 10 to 20 | 69 | This study (09/2017) |
| 204.5-214.5 | Morgat | pXRF | 1 to 10 | 194 | This study (09/2017) |
| 324.6-338.6 | Kerarmor | Lithology | ~1 | 1371 | This study (04/2016) |
| 324.6-338.6 | Kerarmor | NGR | 10 to 20 | 107 | This study (04/2016) |
| 324.6-338.6 | Kerarmor | pXRF | 1 to 10 | 335 | This study (04/2016) |

The gamma-ray data were obtained using a portable spectrometer RS-230 (Radiation Solutions, Inc., Canada). The same instrument was used by Dabard et al. (2015). The measurements were taken with a stratigraphic interval of approximately 10-20 cm and a counting time of 120 seconds. The counts per minute in the selected energy windows are converted to concentrations of K (%), U (ppm) and Th (ppm). The pXRF measurements were done using a Bruker Tracer IV Hand Held portable XRF device equipped with a 2WRh anode X-ray tube and a 10mm2 Silicon Drift Detector with a resolution of 145 eV (Mn-Ka). The X-ray beam is focused on a 6 mm by 8 mm integrated area, using a Pd collimator. X-ray spectra from the pXRF are deconvoluted and quantified using the standard factory "Soil Fundamental Parameters" method. With this technical setting, we could not reliably measure Si, as in the "Soil Fundamental Parameters" method the $SiO_2$ mass fraction is used to sum the total weight to 100% (e.g. explained in more detail in Sinnesael et al., 2018). The fundamental parameters method makes use of the theoretical relationship between X-ray fluorescence and material composition as determined by Sherman (1956). The factory-calibrated quantification method of the pXRF uses this fundamental principle with a correction based on the matrix effect observed in soil and rock samples. All pXRF measurements are carried out by putting the pXRF nozzle directly on the dry and clean outcrop surface. All analyses had a measurement time of 45 seconds (de Winter et al., 2017). The pXRF measurements were not calibrated using external reference materials. As such it is more relevant to consider relative variations in the elemental concentrations data than interpreting absolute concentrations (Sinnesael et al., 2018). As a general principle the pXRF is also less suitable to measure very light (<Al) or heavy elements (>Ba); or elements that only occur in trace amounts – also depending on the instrument used (Beckhoff et al., 2006).

The sequence stratigraphic interpretations follow the method outlined in Dabard et al. (2015) and are based on the study of depositional sequences at different frequencies in combination with NGR measurements (e.g. Fig. 2). There is a variety of sedimentary and condensational facies that can be encountered from the upper part of the Grès Armoricain Formation, throughout the Postolonnec Fm., into the Kermeur Formation (Fig. 1B). Despite their variability, all facies can be ascribed to depositional environments of a terrigenous platform dominated by storms and tides (Guillocheau, 1983; Guillocheau and

Hoffert, 1988; Loi et al., 1999; Botquelen et al., 2006; Dabard et al., 2007; 2015; Pistis et al., 2008; Vidal et al., 2011a,b). Analysis of sedimentary and condensation facies allowed the identification of genetic sequences by identifying the Maximum Regressive Surface (MRS) and Maximum Flooding Surface (MFS) for all sequences of the different hierarchical orders (Catuneanu et al., 2009; Dabard et al., 2015). Due to the high lithological contrast of the sedimentary facies in the upper offshore marine environment, it is easy to identify the MRS and MFS, and its corresponding genetic units. In contrast, in sedimentary environments where facies are homogeneous, such as the shoreface and lower offshore, sequence recognition was mainly based on the identification of condensation facies. In deeper environments, clayey successions are monotonous and condensation facies, such as diagenetic nodular concretionary levels (Loi et al., 1999; Loi and Dabard, 1999; Dabard et al., 2007; Dabard and Loi, 2012; Loi and Dabard, 2002) or shellbeds levels (Botquelen et al, 2004; 2006), are the main markers that allow effective sequential partitioning especially when coupled with NGR measurements of U versus Th. In homolithic sandy shoreface facies, condensation levels have been detected through NGR analysis. Levels that show considerable increases in NGR U and Th contents correspond with facies of heavy mineral accumulations (paleoplacers) and are interpreted as an expression of condensation (Pistis et al.,2008; 2016; 2018).

Numerical spectral analyses were carried out using the available pXRF and NGR data as summarized in Table 1. Stratigraphic intervals that correspond with 'event beds' (e.g. turbidites, volcanic ashes, small slumps) are sometimes removed from time-series analysis as they are not considered to be representative of the environmental sedimentary signal of interest (e.g. Zeeden et al., 2013). Although sandstones have relative higher accumulation rates compared to mudstones, we do not exclude the sandstone intervals from spectral analysis in this study. Here, also the mudstones are non-continuous (on short timescales) storm deposits, and the variation of K reflects the changing mineralogy which is controlled by grainsize. Besides the fact that some intervals like the Kerarmor Mb. consist mostly of sandstones, removing the sandy intervals would remove the main indicator of bathymetric variation which is the main paleoenvironmental variable for this study. Evolutive harmonic analysis (EHA) (Thomson, 1982), bandpass filtering and TimeOpt (Meyers, 2015, 2019) analyses were done with "Astrochron" (https://cran.r-project.org/web/packages/astrochron/index.html; Meyers, 2014) in R (R Core Team, 2021). TimeOpt is a statistical optimization method that can simultaneously consider power spectra distributions and precession-eccentricity amplitude modulation patterns (Meyers, 2015, 2019). Data were linearly interpolated and linearly detrended unless specified otherwise. All specifications regarding multi-taper settings, sliding window sizes, bandpass filters and TimeOpt parameters can be found in the supplemented R script (Supplement).

## 4 Stratigraphic and geochemical results

The first type of data contains the facies analyses considering both sedimentary structures and variations in grain size (Fig. 4). The Kerarmor Mb. is characterized by a much larger stratonomic variation in grainsize throughout the section than the Morgat Mb., with a large range of facies corresponding with various paleoenvironments, which makes it more

straightforward to differentiate the different depositional sequences related to several orders of relative sea-level change. The Morgat Mb. essentially consists of a continuous silty mudstone facies, overlying the top of a thick sandstone unit of the upper Kerarvail Mb. The more homogeneous distribution of lithological compositions makes facies analysis and high-frequency interpretation of possible changes of relative sea level more difficult. In the lower offshore (<~120 m water depth), more clayey successions Loi and Dabard (2002) associate facies with slight bioturbation and storm laminae with periods of relatively higher sedimentation rates (falling sea level), and the more bioturbated facies that contains the nodules with periods of relative slower accumulation rates (rising sea level). During these times of low sedimentation rates and nodule formation (early diagenetic), it is possible that several very high frequency sequences are amalgamated, indicating that even in the more mudstone-dominated intervals changes in sedimentation rates can be pronounced. Unfortunately, we only measured individual nodules with the pXRF for the Morgat Mb. Section, which was visited after measuring the Kerarmor Mb. where nodules are a much less prominent feature. Individual nodules measured with the pXRF for the Morgat section are indicated by the full circles in Fig. 4, while measurements on shell beds are indicated by triangles. As the NGR device measures natural radioactivity within a radius of at least 10 cm, it is not possible to analyze individual nodules, which have a diameter ranging from a few mm up to a few cm. The sequence stratigraphic analysis based on the facies interpretation shows a first curve at high frequency reflecting variations in sedimentary environments (purple line in Fig. 4). We attribute this curve to relative sea-level variations, which is the main signal that can preserve different sedimentary facies. The low frequency curves (red and blue lines in Fig. 4) are obtained by smoothing the higher-frequency curve and do not correspond point by point to the sedimentary environments, but to the lower frequency signal that modulates the higher frequency. Astronomical cycles mentioned in Fig. 4 are only suggestive, *sensu* Dabard et al. (2015). It is evident that the thicknesses of the sequences of the same order vary enormously according to the environments in which they developed. In the proximal environments, they are dilated while towards the lower offshore environments they appear less thick due to the decrease in terrigenous input caused by the increased distance of terrigenous sediment transport. Usually, more distal successions feature also much less event beds.

For both high-resolution stratigraphic logs the first-degree variations in lithology are well reflected in the K curves measured by both pXRF and NGR (Fig. 4). The mudstone intervals have a larger clay content and correspond with higher K values and vice versa for the sandstones. There is also an excellent agreement between the low-resolution NGR data published in Dabard et al. (2015) and the new high resolution NGR data (Supplementary Materials). While the relative variations in K for both the pXRF and NGR agree well, there is an offset in the reported absolute concentrations because of the lack of absolute concentration calibration for the pXRF measurements. The same pattern in mudstone versus sandstones is also visible in the pXRF Rb profile, which again follows the clay content (Supplementary Materials). For both stratigraphic profiles, variations in NGR U and Th concentrations behave overall similarly and seem to follow the first-degree features of the NGR K record. However, the relative variations in NGR U and Th concentrations are less pronounced in amplitude and show a less consistent relationship with the measured lithology. For example, while U and Th generally are expected to be enriched in

shales and clays (Adams & Weaver, 1958), the thick sandstone unit on the top of the Kerarvail Mb. has the highest measured U and Th values (Fig. 4), due to the presence of paleoplacers. The term 'paleoplacer' refers to heavy mineral-rich laminae that often occur on the top of thick sandstone units of the Postolonnec Fm. and other similar paleoenvironmental depositionary settings (Pistis et al., 2008; 2016; 2018). Also, the thick sandstones (>12 m) at the top of the Kerarmor Mb. have elevated U and Th values (with paleoplacers), while the sandstone unit around 11 m has low U and Th values (without paleoplacers).

The pXRF measurements also yielded two other groups of elements, which variations in concentrations seem to be driven by sedimentological or diagenetic processes, i.e., in being associated with paleoplacers and concretions (nodules). A first group of elements measured with the pXRF relates to paleoplacers (Zr, Ce and Ti, Fig. 4). An example of such a paleoplacer-rich stratigraphic interval is the top of the thick Kerarvail Mb. sandstone unit, which is characterized by high levels of natural radioactivity due to the presence of certain minerals. Certain elements associated with this mineral assemblage (e.g. zircons, monazite and titaniferous minerals as described by Pistis et al., 2016; 2018) can be detected by the pXRF. This is most clearly reflected in the Zr and Ti profiles. Our 'single spot per bed' measurements do not allow for specific mineralogical identifications, but an approach that combines multiple pXRF measurements for the same sample for coarse-grained igneous rocks has shown promising results to extract mineralogical information (Triantafyllou et al., 2021). In theory, Ce should also show elevated values in the paleoplacers, but this element is potentially too heavy (and can occur in too low concentrations) to be reliably detected by the pXRF analysis. Some elements like Ti occur in elevated concentrations in both paleoplacers and other facies. Considering both NGR and pXRF data together can be particularly useful. For example, combined elevated concentrations of U (NGR), Th (NGR), Zr (pXRF) and Ti (pXRF) are a very strong indicator of a paleoplacer. Around 6.5 m in the Kerarmor section there is an elevated Ti (and Ce) peak for measurements performed at the base of a thick sandstone (Fig. 4). This level does, however, not show higher concentrations of Th, U or Zr. Possibly the small spot of the pXRF hit a specific mineralogy, or represents 'an outlier'. All datapoints measured in the field are reported in Fig. 4, without arbitrary screening for potential 'outliers'. The pXRF elemental analysis can thus be a fast and useful tool to perform multi-elemental analysis in the field and detect potential paleoplacers.

Various types and forms of concretions can be found throughout the Postolonnec Fm. (e.g. Loi and Dabard, 2002; Dabard and Loi, 2012). The most common concretions are cemented by silica, phosphorus or carbonate. Unfortunately, due to the used technical settings of the pXRF measurements Si was not reliably reported (see Methods). Nevertheless, the pXRF analysis still reveals useful information. Some concretions in the Morgat Mb. 8-10 m interval have high Ca values and are most probably carbonate cemented concretions (Fig. 4). Some measured shell beds also show elevated Ca values reflecting a preserved fraction of carbonate mineralogy of the original shells. These Ca-enriched nodules have also very low concentrations of K, Rb and Ti. Another group of concretions have elevated Ce values (although caution is required with respect to the quality of the measurement for this element) and K, Rb and Ti values, which are similar to their surrounding

sediments. This group corresponds with the phosphatic concretions as Ce is one of the elements, which is typically enriched in the phosphatic apatite. The nodules can also be enriched in U (NGR), where it is linked to the presence of organic matter, and at the same time often display low Th (NGR) concentrations. More systematic work is needed, where the same nodules are analyzed with both pXRF and more robust analytical techniques (e.g. Inductively Coupled Plasma Mass Spectrometry), but these first results demonstrate the potential as a useful non-destructive screening tool usable in the field.

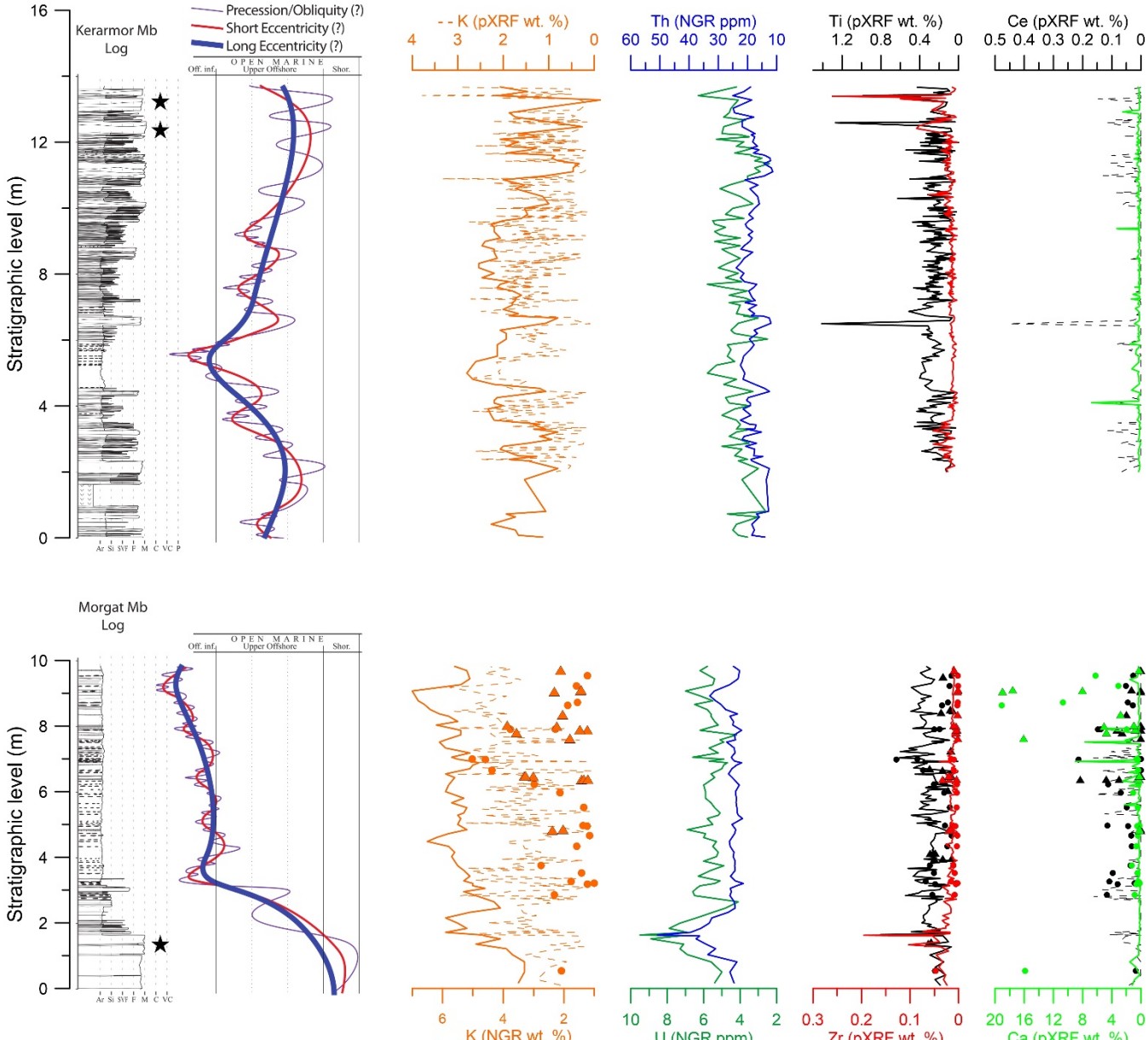

**Figure 4: Sequence stratigraphy and geochemical results from the portable X-ray fluorescence (pXRF) and natural gamma-ray (NGR) measurements. Several orders of sea-level change are interpreted in a sequence stratigraphic framework (very high frequency (purple and red curves), high frequency or fourth-order (blue curve). The association with respective possible**

**astronomical cycles of precession, obliquity or eccentricity is only suggestive,** *sensu* **Dabard et al. (2015). Note that the pXRF concentrations are reported in weight percent (wt. %) as measured by the internal instrumental calibration (Soil Method), which in the absence of an additional external calibration are not accurate. The pXRF measurements are more meaningfully interpreted in function of relative changes. The black stars indicate the stratigraphic occurrence of paleoplacers.**

## 5 Cyclostratigraphic analysis

The cyclicity analysis of Dabard et al. (2015) was based on the identification of several orders of sea-level fluctuations. The most pronounced order of sea-level variations is the meter-to-decameter-thick 'high-frequency cycle' (Fig. 2). By taking the ratio between the number of identified high-frequency cycles (n=36) and the GTS2012 estimated duration for the Darriwilian (8.9 Myr, Cooper and Sadler, 2012; Cohen et al., 2013) a maximal duration estimate of 410 kyr per cycle was obtained. Therefore, this cycle was ascribed to the ~405 kyr long eccentricity cycle, which is theoretically the most stable astronomical

cycle that is also recorded throughout the Phanerozoic (Laskar, 2020). The higher order frequency cycles were then respectively ascribed to the ~100 kyr eccentricity and ~20 kyr precession cycles. The absence of more precise temporal constraints, other than chitinozoan biostratigraphy, prevents further robust testing of this hypothesis. Furthermore, it makes it difficult to clearly distinguish a potential short-term Ordovician precession cycle (~20 kyr) from an obliquity (~30 kyr) cycle (Berger and Loutre, 1994; Laskar et al., 2004; Waltham, 2015). Considering a potential obliquity cycle is especially relevant

given that the Darriwilian is also hypothesized to have featured some of the earliest Ordovician glacial episodes (e.g. Vandenbroucke et al., 2010; Dabard et al., 2015; Pohl et al., 2016; Rasmussen et al., 2016), with high-latitude glacial dynamics typically being more sensitive to an obliquity forcing (Hinnov, 2018). The long-term 1.2-Myr obliquity cycle has been hypothesized to play a role in the Ordovician icehouse stratigraphic structures (Turner et al., 2012; Ghienne et al., 2014; Dabard et al., 2015). An important nuance is that the so-called very long 2.4-Myr eccentricity and 1.2-Myr obliquity

cycles do not necessarily have the same duration during the Ordovician due to the chaotic behavior of the solar system (Laskar, 1989; Olsen et al., 2019; Hoang et al., 2021). The highly variable sedimentation rates and potential (small) hiatuses also make it challenging to test for potential amplitude-modulation patterns that could identify a unique astronomical imprint throughout the entire stratigraphy (Meyers, 2015; 2019; Sinnesael et al., 2021). A further challenge to consider is that the Ordovician geological time scale is continuously evolving, with the most recent duration estimates for the Darriwilian being

11.2 Myr (GTS2020 graptolite composite) or 11.6 Myr (GTS2020 conodont composite) versus the GTS2012 much shorter 8.9 Myr estimation (Goldman et al., 2020). These duration estimates vary up to 30 %, which might not change that much to the order of magnitude of the cycle duration estimates, but certainly has implications for the potential completeness of the record. Moreover, one has to consider the difficulty of precisely identifying the stage boundaries based on chitinozoan biozonation solely. A last limitation relates to the difficulty to clearly distinguish stacked patterns in the very condensed

parts of the Postolonnec Fm. (Dabard et al., 2015). Rather than suggesting a number of cycles that agrees with certain stage duration estimates, there is much more value in trying to identify the actual number of cycles clearly distinguishable and considering the possibility of missing cycles. Especially in the shallow marine shelf environments, it is reasonable to for example consider periods of non-deposition or erosion. Although many of the discussed challenges are inherent to the nature

of the record and lack of independent constraints, we consider two additional elements to the cyclostratigraphic analysis of the Postolonnec Fm.: (i) investigate if new high-resolution pXRF data in the highly condensed interval (Morgat Mb.) are complementary to the NGR data and can help with clearly distinguishing sequences and (ii) if the use of traditional spectral analysis tools is valid and of added value.

To investigate the most condensed part of the Morgat Mb. high-resolution profile, we focus on the interval that does not contain any silt or sandy layers (from 3.5 m above the base of the profile till the top at 10.0 m). The main sedimentological information that could discriminate potential sequences here are the occurrence of shellbeds, nodules (Loi and Dabard, 2002) or variations in elemental concentrations as measured by NGR and pXRF.

Let us first consider the medium-resolution NGR K signal (Fig. 5B). The signal shows some fluctuations in concentrations varying between 5 and 6.5 %K. Here, we use evolutive harmonic analysis (EHA) to evaluate the spectrum of the signal as it evolves throughout the stratigraphy (Thompson, 1982). The frequencies that explain more variation within the moving window will have higher spectral power and are shown by redder colors (Fig. 5B). This approach has the advantage over a single periodogram or multi-taper spectrum that it can also be used to evaluate how stratigraphically consistently present a certain period might be or not. Spectral analyses indicate two main periodicities: a longer one of ~1.5-2.0 m (0.5-1.0 cycles/m) and a shorter one around ~0.5 m (1.8-2.2 cycles/m) cycle thickness (as indicated by the dotted lines on Fig. 5A). The rest of the EHA shows little elevated spectral power for other frequencies. The data set is relatively short (6.5 m) with only 46 data points and an average sample resolution of 14 cm, so one must consider that there are only few of the 1.5-2.0 m cycles in the record, and the 0.5 m cycle is close to the theoretical Nyquist frequency (stating that you need at least two data points per cycle to be able to detect it, in practice it is often better to have even more (e.g. Martinez et al., 2016)). However, the bandpass filters of both periods show a good agreement with the raw signal (Fig. 5C). Let us now consider the pXRF K signal that has a higher average sampling resolution (6 cm) and more data points (n=106) for the same investigated stratigraphic length (Fig. 5E). The spectral analysis of the pXRF K signal shows again the ±1.5 m cycle, a less pronounced 0.5 m cycle and new dominant cyclicity around 0.25 m (Fig. 5F). Compared to the NGR EHA, the pXRF EHA suggests additional frequencies with lower spectral power. These seem, however, less stratigraphically continuous (Fig. 5). The bandpass filters of the NGR and pXRF signals demonstrate a similar number of cycles, but they are not perfectly in phase with each other (Fig. 5D). The pXRF 0.25 m periodicity bandpass filter shows a pronounced amplitude modulation, which is consistent with the amplitude modulation in the 1.5 m periodicity for the pXRF signal– although again not perfectly in phase (Fig. 5C-D). The pattern of the amplitude modulation in combination with the number of shorter cycles (5-6) in the longer cycles is suggestive of a precession-eccentricity signal. This signal is consistent with the ratio fitting of the 1.5, 0.5 and 0.25 m periodicities to the predicted Ordovician duration of the astronomical cycles of short eccentricity, obliquity and precession (Berger and Loutre, 1994; Laskar et al., 2004; Waltham, 2015), as well as with the order of magnitude estimate sedimentation rates estimated by Dabard et al. (2015) for this interval. In fact, it is of interest that Dabard et al. (2015),

independently of this study, interpreted 405-kyr eccentricity minima roughly at the same stratigraphic positions as we identify small amplitude variations in a potential precession signal. A precession-eccentricity signal can also be statistically tested with the TimeOpt approach (Meyers, 2015, 2019). The TimeOpt results are close to our interpretation with the nominal TimeOpt outcome (assuming constant sedimentation rate over the whole record) that suggests the presence of 5 short eccentricity cycles or a sedimentation rate of 12 m/Myr; while the evolutionary eTimeOpt analysis suggests slightly higher sedimentation rates (15 m/Myr) with a small increase in sedimentation rate up section (R script for TimeOpt analyses provided in the Supplement). It is hard to further demonstrate an astronomical origin of these variations in the absence of more precise stratigraphic constraints. However, our observations based on a numerical approach using high-resolution geochemical data were consistent with those of Dabard et al. (2015), suggesting our new approach can be an additional useful tool. The occurrences of the shell beds (mainly between 6.5 and 9.5 m) and nodules are not so clearly cyclically distributed, making it difficult to robustly assess their usability as tool for the assessment of 'condensed sequences'. That nodules occur less systematically might be related to the interplay of various boundary conditions that can lead to a varied sedimentological expression of a condensation interval (as discussed in Loi and Dabard (2002) and Dabard and Loi (2012)).

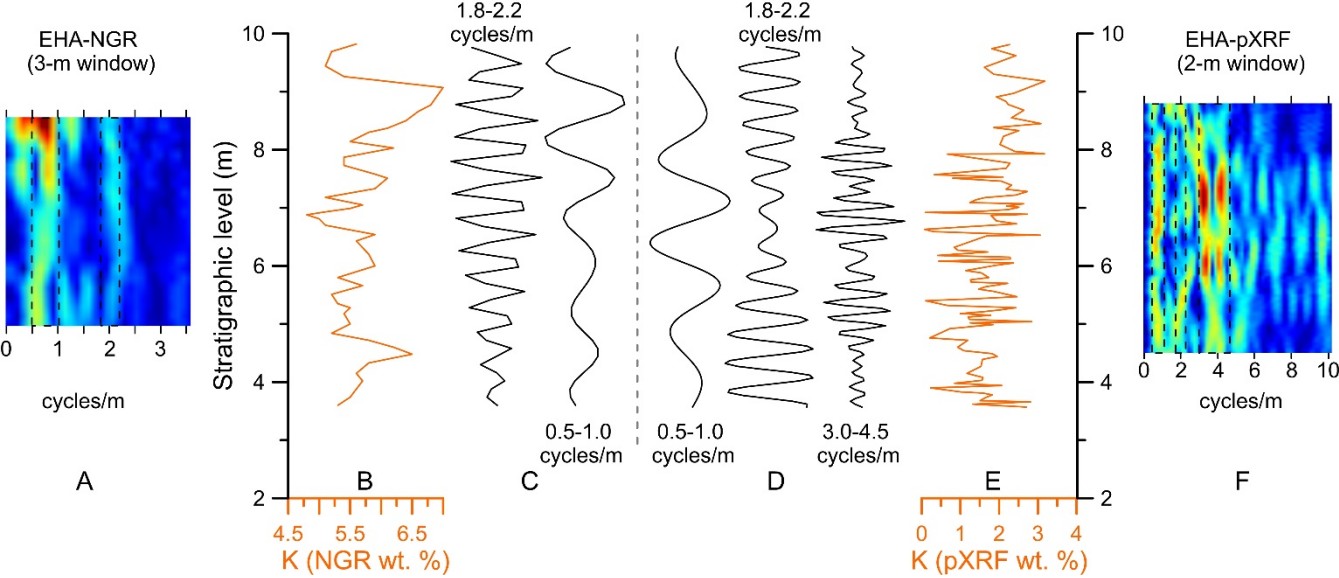

**Figure 5: Spectral analysis of the Morgat Member potassium records showing the measured data (B&E, orange lines), several bandpass filters (C&D, black lines) and evolutive harmonic analysis (A, F, spectral plots with ranges of bandpass filters indicated by dotted rectangles). Left panels are based on the natural gamma-ray record (A,B&C) and the right panels are based on the portable X-ray fluorescence data (D,E&F). EHA = evolutive harmonic analysis.**

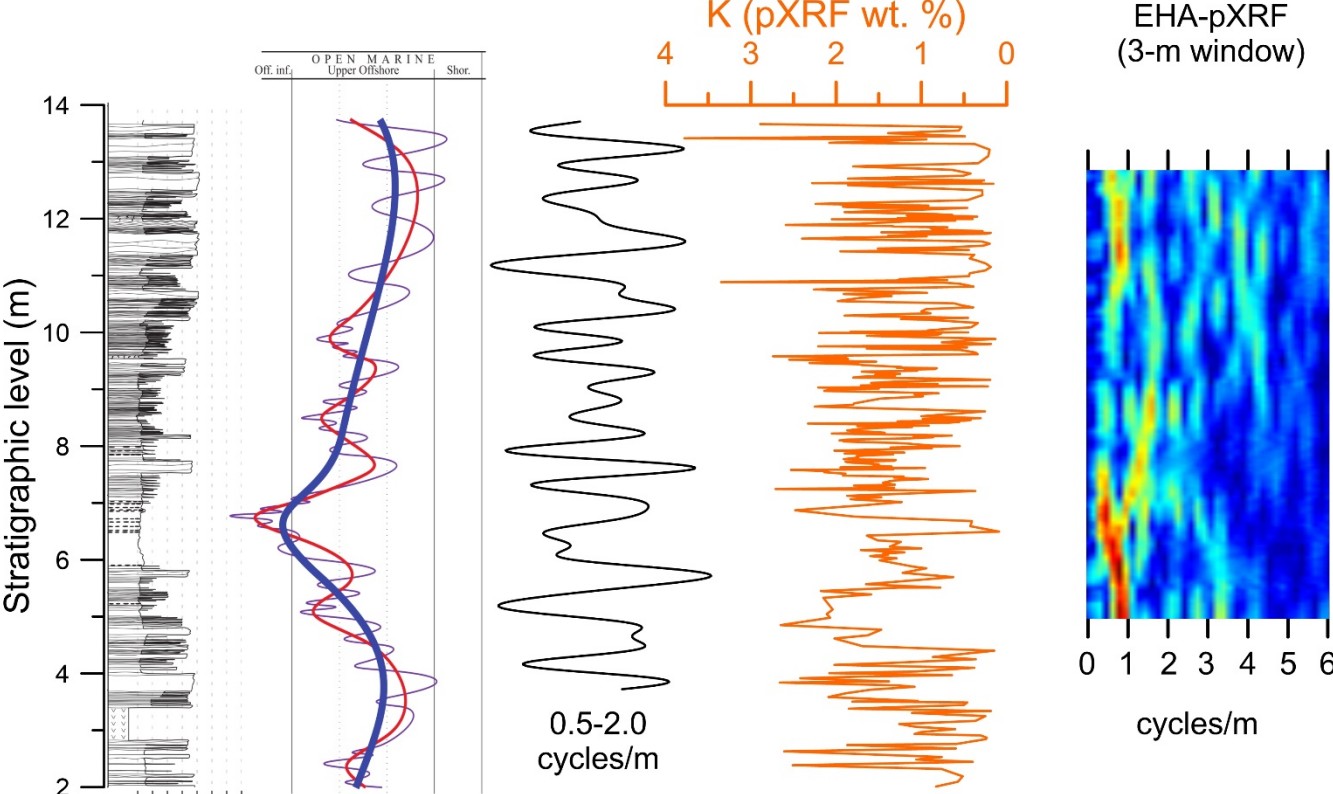

**Figure 6: Comparison of sequence and numerical cyclostratigraphic analyses for the Kerarmor Member based on the facies, potassium from portable X-ray fluorescence and bandpass filtering. EHA = evolutive harmonic analysis. Several orders of sea-level change are interpreted in a sequence stratigraphic framework (very high frequency (purple and red curves), high frequency or fourth-order (blue curve).**

We can now apply a similar approach to the pXRF K record of the shallower marine Kerarmor section that is characterized by a larger variation in facies. The power spectrum is dominated by a ~0.5-2.0 m (0.5-2.0 cycles/m) cyclicity, apparently with thicker cycles in the more sandstone dominated intervals (below 5 m and above 10 m) and thinner cycles within the more clay-and siltstone facies. One logical interpretation could be that the same dominant sequences get thinner within the finer fractions. The bandpass filter of this frequency range does not show any pronounced amplitude modulation. The bandpass filter output does show some resemblances with the independently constructed relative sea-level reconstructions. The thick sandstone beds are identified by both methods as individual cycles, while they probably represent a single (or few) depository event(s). The sequence stratigraphic interpretation suggests the presence of a larger number of high-frequency cycles in the finer fractions compared to the bandpass approach that is limited by its user-defined frequency width. In the Dabard et al. (2015) interpretation the highest frequency relative sea-level changes were interpreted to correspond with the precession (or obliquity) cycles, and smoothed higher order cycles with eccentricity cycles. Again, in the absence of better

age control this hypothesis is difficult to test further. The numerical cyclostratigraphic analyses did not provide further indications of a potential astronomical origin of these sequences but did offer another way of describing the variations within the signal.

The detailed study of these two short sections indicates that the commonly used numerical cyclostratigraphical approach may work for the selected short and relatively lithologically homogenous interval, but might be challenged when applied to intervals with more contrasting lithologies. To test this idea further, we investigate the spectral analysis properties of the whole Postolonnec Fm. (11-393 m on the depth scale of Dabard et al. (2015)), based on the available low-resolution (m-scale) NGR data. The importance of sampling resolution to potentially pick up smaller-scale cycles was already illustrated by the difference in outcomes for the NGR versus pXRF data for the Morgat Mb. analysis. With an average sample rate of ~0.5 m, it might be expected that shorter-term astronomical cycles such as precession or obliquity cannot be recovered by numerical analysis of the low-resolution NGR record. Although the expression of such potential cycles in a relatively coarse siliciclastic facies may be very different (e.g. they might be much thicker and detectable with a relative low sampling rate). A first complication is that the amplitude of the variations of the NGR signal is much larger in the sandstone-rich intervals (i.e. Kerarvail Mb. and Kerarmor Mb.) compared to the lithologically more homogenous intervals, which makes the spectral power much higher in the sandstone intervals, masking potential smaller amplitude changes in the other intervals (Fig. 7). An additional complication is that the very variable lithological succession results in quite a complex long-term trend in the NGR signal that can not be fully removed by subtracting a simple linear trend (the most common and basic form of detrending a signal prior to spectral analysis). It is possible to perform more complex ways of detrending, for instance by using a polynomial trend or lowpass filtering. The challenge is finding the delicate balance between removing what is long-term trend and the signal to recover. Due to the potential large variation in sedimentation rates (up to an order of magnitude) and paleoenvironments, this is a difficult balance to strike for this dataset. We have applied various levels of detrending using a lowpass filter to assess the robustness of the consecutive spectral analyses (evaluated between f= 0.01-0.05 m$^{-1}$, or periods > 100-20 m). In Fig. 7 we compare the EHA's for a simple linear detrending and a lowpass filtered detrending (f= 0.033 m$^{-1}$, or periods > 30 m). The detrending has decreased the difference in spectral power between different lithologies, but the overall pattern stays the same. Both the Kerarvail and Kerarmor sandstone members show a consistent 4 and 2.5 m cycle. The longer-term cycles are less robust to identify and depend partially on the level of detrending. Applying the respective sedimentation rates of the Dabard et al. (2015) interpretations for these stratigraphic intervals would result in a ~100 kyr (short eccentricity) duration for the 4-2.5 m cycles. Interestingly, sometimes a 12 m thick cycle (~400 kyr) also appears in the spectral analyses results, but only in the most heavily detrended signals. Overall, this is a more tentative and not fully independent interpretation, as Dabard et al. (2015) made their interpretations assuming an astronomical origin of their signal. Striking the balance between removing long-term trends and what is a potential astronomical signal proofs to be particularly challenging for this long record with a pronounced change in sedimentary environment. One could additionally analyze every subsection individually, for example with high-resolution pXRF data, but the absence of independent age

controls remains. In the absence of datable volcanic ash beds one way forward to obtain some numerical age constraints could be the dating of detrital zircons. Even though detrital zircons do not give a depositional age, recent technological advances make it possible to for example date a large number of detrital zircons (with laser-ablation–inductively coupled plasma–mass spectrometry, LA-ICP-MS) after which the youngest zircons can be very precisely dated (with chemical

abrasion–isotope dilution–thermal ionization mass spectrometry, CA-ID-TIMS) to come to an informative 'maximal depositional age' (e.g. Karlstrom et al., 2019; Landing et al., 2021). A conceptually related approach concerns the dating of prismatic zircons in a Darriwilian limestone bed in Sweden (Lindskog et al., 2017; Liao et al., 2020). For both the detrital zircon and non-bentonite associated prismatic zircon dating approaches it is crucial to keep in mind that the numerical age from the dating comes with an additional (larger) uncertainty on its depositionary age. Even when such uncertainties would

be in the order of millions of years, they can still be valuable in such cases where there are tens of millions of years worth of little chronometrically constrained stratigraphy.

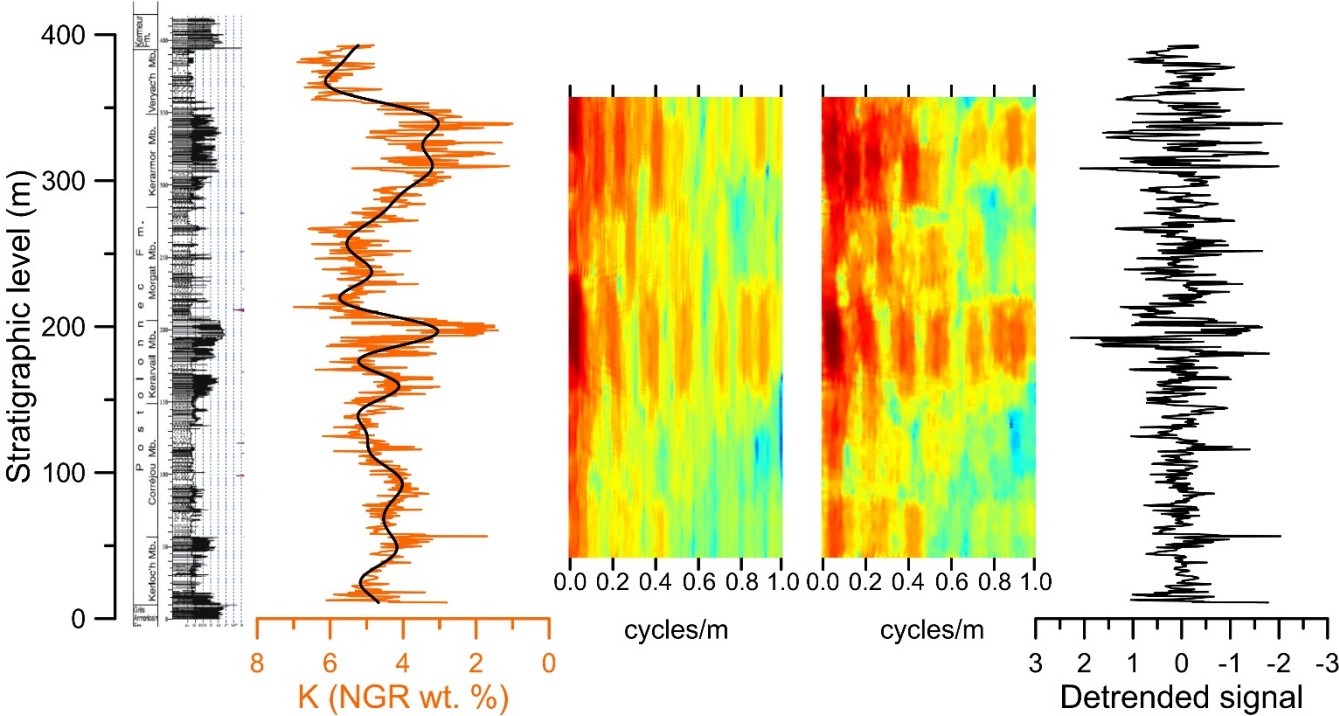

**Figure 7: Spectral analysis results based on the potassium natural gamma ray record for the whole Postolonnec Formation**
**demonstrating the influence of detrending.**

Alternatively, the "spectral moments" approach is designed to pick up potential first-order changes in pronounced sedimentation rate changes in astronomically forced signals (Sinnesael et al., 2018b). The main underlying idea of the approach is that some of the main characteristics of the evolving spectra (e.g. mean frequency and bandwidth) shift frequencies with changing sedimentation rates. For a same astronomical imprint under lower sedimentation rates

(corresponding to a sedimentary sequence evolving into more distal sedimentary environments), the same cycles are respectively less thick and corresponding frequencies are higher. In general, there is also a higher occurrence rate of event beds in more proximal and sandstone-dominated intervals, which are most probably not astronomically influenced. In our case, the appearance of additional higher frequency cycles in the sandstone intervals of the Postolonnec Fm. shifts the spectral characteristic values to higher values, implying lower sedimentation rates. This is the opposite signal than expected

(higher sedimentation rates for sandstone intervals compared to mudstone intervals) and can be explained by the extra components that appear in the higher frequency ranges (e.g. around 0.55, 0.7 and 0.9 cycles/m in Fig. 7) compared to the more condensed intervals, rather than changes in sedimentation rate. It is important to consider that the sedimentation rate of sandstone is very likely to be higher than condensed mudstones, but sandstone intervals equally represent a less complete or continuous sedimentary record. Therefore, a purely numerical approach to formulate for example a duration estimate for the

studied section might be invalid and not necessarily more robust than the more qualitative approach of Dabard et al. (2015) which is sometimes also applied in similar studies (e.g. Loi et al., 2010). Another consideration is indicating a degree of (un-)certainty: some intervals might show clearly distinguishable sedimentary cycles while others are less clearly interpretable. By explicitly formulating these types of uncertainties one can inform potential future studies and come to more informative duration estimations (Cramer et al., 2015; Sinnesael et al., 2019).

Formulating cyclostratigraphic uncertainties in a Paleozoic integrated stratigraphic framework is not an easy task (e.g. Sinnesael et al., 2019; Ghobadi Pour et al., 2020). Studies that, similarly, target less conventional facies in younger stratigraphical intervals might in general have more robust independent age constraints (e.g. Noorbergen et al., 2018) or more reliable astronomical parameters like insolation curves available (e.g. Vaucher et al., 2021). This is much less the case

for the Paleozoic (e.g. Laskar, 2020), often resulting in looser temporal constraints on astronomical interpretations. For example, Sinnesael et al. (2021) reinterpreted the expression of astronomically forced Upper Ordovician sedimentary cycles on Anticosti Island (Long, 2007; Elrick et al., 2013) resulting in a different interpretation of the duration of the cycles by an order of magnitude. The use of correlations and ages that only are loosely constrained, in order to imply astronomical origins of sedimentary sequences, is not uncommon when interpreting lower Paleozoic records (e.g. Sutcliffe et al., 2000;

Gambacorta et al., 2018). Other common practice is the application of spectral techniques on stratigraphic records that might not be ideal for such type of analysis because of, e.g., their variable lithologies and associated variable expression of the proxies used (e.g. Zhong et al., 2018). These challenges accentuate the need for tailored cyclostratigraphic methodologies that are not simply a copy of what has been shown to work well for younger stratigraphic intervals; instead we need techniques that are adapted to the reality of both the more limited availability of accurate independent age constraints and the

absence of well-preserved open marine pelagic sections that characterize the Paleozoic sedimentary record.

**6 Conclusions**

Using pXRF measurements directly on outcrops can be challenging because of superficial weathering of the rocks and the need for flat measurement surfaces for XRF. The protected Postolonnec beach section is characterized by strong tidal ranges and coastal erosion. This creates fresh and smooth surfaces but that also are prone to be covered by biological material. Despite the challenging nature of the outcrop, we demonstrate that the non-destructive pXRF measurements trace lithological changes and several crucial sedimentary features such as the occurrence of paleoplacers or condensation horizons, as reflected by the presence of different types of diagenetic concretions such as nodules.

The relative variations in the pXRF K and NGR K are very similar in the measured logs. A comparison with the NGR high-resolution logs of this study and the low-resolution logs presented in Dabard et al. (2015) shows that these NGR measurements are robust and reproducible. An important reason for this reproducibility is the fact that the NGR measurements average out natural radioactivity within a radius of at least 10 cm. pXRF analyses have a ~1 cm spot size. This allows for a much higher sampling rate, the measurement of individual nodules or paleoplacer beds, but this also means that the measurements pick up more detailed small-scale variations and are harder to reproduce over different field visits. Another advantage of the pXRF is the multi-elemental nature of the analysis.

The use of high-resolution pXRF measurements and traditional cyclostratigraphic tools delivered promising results to distinguish sedimentary cycles in the relatively homogeneous condensed mudstone facies. Application of commonly used spectral analysis tools for the whole record with strongly varying facies is shown to be challenging, with a particular difficulty of distinguishing longer-term trends and potential astronomical signals. In the absence of precise independent age constraints, it is not possible to fully confirm the astrochronological framework suggested in Dabard et al. (2015). A potential astronomical signal is suggested in the more homogenous mudstone facies, while reliable cycle identification in the more proximal sandstone-dominated intervals proofs to be challenging. More work is needed to expand our toolboxes to also study non-traditional sedimentary archives in terms of astronomical climate forcing to advance our knowledge of deep-time cyclostratigraphy.

**Code and Data availability**

All produced geochemical data, sedimentary logs and R-code used for spectral analyses are provided in the Supplement.

## Author contribution

MS, TRAV and PC conceptualized the research. MS, AL and MPD performed the field surveys and geochemical measurements. AL performed sequence stratigraphic analyses. MS performed spectral analyses. MS wrote the original manuscript with contributions from all co-authors to the revised drafts.

## Competing interests

The authors declare that they have no conflict of interest.

## Acknowledgements

We would like to dedicate this study to Marie-Pierre whose knowledge and passion for the Postolonnec was unprecedented, and was taken away too soon. Matthias Sinnesael thanks the Research Foundation – Flanders (PhD fellowship FWOTM782) and is funded by the European Research Council (ERC) under the European Union's Horizon 2020 research and innovation program (Advanced Grant AstroGeo-885250). This work was supported by the Fondazione Banco di Sardegna [grant numbers F74I19000960007 and F75F21001270007]. Thijs Vandenbroucke thanks the The King Baudouin Foundation (Professor T. Van Autenboer Fund) and the Bijzonder Onderzoeksfonds (BOF- UGent-BOF17/STA/013) for funding. Philippe Claeys thanks the VUB Strategic Research Program for funding and the FWO Hercules foundation for financing the XRF analytical platform at the VUB. This work contributes to International Geoscience Programme projects IGCP 652 (Reading geologic time in Paleozoic sedimentary rocks) and IGCP 753 (Rocks and the Rise of Ordovician Life).

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
