# Peer review of "Cyclostratigraphy of the Middle to Upper Ordovician successions of the Armorican Massif (western France) using portable X-ray fluorescence"

_Geochronology, 2021_

## Referee Comment (RC1)

**Review Sinnesael et al. Cyclostratigraphy of the Middle to Upper Ordovician successions of the Armorican Massif (western France) using portable X-ray fluorescence**

The manuscript of Sinnesael et al. present new portable XRF data obtained from an Ordovician succession from western Brittany (France). The sequence stratigraphy was previously published. The succession corresponds to the record of an open marine environment, with deposits covering from the shoreface to the lower offshore. This change in environment trigger changes in type of sediment deposited (sand vs. clay) but also change in sediment rates, with higher sediment rates in more proximal deposits.

I understand from the manuscript that the section is protected and no sampling was performed because of this reason. Measurements were thus performed *in situ* using spectral gamma ray and portable XRF (pXRF). Spectral analyses were performed from the potassium content approximated from the spectral gamma ray and pXRF. Two members were analyzed: the Morgat Mb., consisting of claystone deposited in a lower offshore environment, and the Kerarmor Mb., representing alternations between claystone and sandstone deposited in an upper offshore to shoreface environment.

From the spectral analyses done on the K series, the authors conclude that the Morgat Mb. Contains an excellent record of the Milankovitch cycles, unlikely to the Kerarmor Mb. The authors attribute the inability of spectral analyses to identify the sedimentary record of the Milankovitch cycles in the Kerarmor Mb. to the highly unstable sedimentation rate, demonstrating that the sequences, the processes and environments at the origin of the deposits have to be identified before applying Fourier analyses, and their derivative methods, on a sedimentary series. To be franc, this conclusion brings little novelty. This is a useful reminder that cyclostratigraphy, before being a statistical challenge, is a geological challenge. If one wants to suggest to have the record of the Milankovitch cycles, they have to prove first that they have climatic and/or eustatic cycles which average period is within the range of the astronomical cycles. From a Fourier analysis, this is possible to find a frequency content that mimics the astronomical cycles, with however no climatic or eustatic significance. Relying only on a Fourier transform done from a geophysical signal is thus not enough, especially if event deposit, such as storm deposits, are included in the sampling and the frequency analysis. This shall be obvious, and one might consider this conclusion as naive. This is however sometimes forgotten in publications.

Saying that, I find the spectral analyses of the Kerarmor Mb. superficial, in a sense that, knowing that the time of deposit of sandstone beds can be regarded as instantaneous compared to the time needed to deposit the claystone beds, the authors should try spectral analysis of the K content on the section "sandstone-free", i.e. removing data and thickness of the sandstone beds from the series to only keep the decantation deposits, which sedimentation rate and variability of K content was probably much more stable. In complement, the authors could use the study of Dabard et al. (2015) to convert the sequences they attribute to the precession cycles to they expected average period (so they make an orbital tuning from the sequencing of Dabard et al., 2015) to remove the variations in the sedimentation rate obviously depending on the sandstone beds only.

In general in this manuscript, I find the graphic representation of the spectra unclear. The description of the spectra is also extremely superficial. I redid the spectral analyses. Below are the spectrum and the power spectrogram of the K content in the Morgat Mb. It appears that the frequency the authors chose (1.5 m, 0.5 m and 0.3 m) are not the highest powers or confidence levels regarding a red noise. Can the authors explain their choice? Is it based on their stratigraphic continuity? What is the origin of the other frequencies?

[Figure]

Figure 1: 2π-MTM spectrum of the pXRF K content from the Morgat Mb. Purple line is median smoothing; brown line 90 % confidence level; green line 95 % CL and pink line 99 % CL.

[Figure]

Figure 2: Evolutive Fourier transform of the pXRF K content from the Morgat Mb. The red lines are spectral peaks. The blue color indicates the spectral background. The Fourier transform were done on 5-m intervals.

In the chapter of the spectral analysis of the Kerarmor Mb., the author apparently experienced difficulties in calculating the long-term trend, which surprised me. In short sections, it may indeed not be trivial, however in this case, applying a locally weighted scatterplot smoothing curve with a coefficient of 0.5 allows the lowest frequencies to be decreased to low values while preserving the spectral peaks at higher frequencies. Notice that with this procedure, no spurious peak is produced at low frequencies, following the recommendation from Vaughan et al. (2011).

[Figure]

Figure 3: Detrending of the pXRF K content of the Kerarmor Mb. Top right figure is the spectrum before detrending (only the average of the series is set to 0 for clarity). Bottom right is the spectrum after detrending applying a LOWESS with a coefficient of 0.5.

In summary, I find the description and the design of the experiment extremely superficial, and additional work is needed in my opinion. So, at the moment I am not convinced by the design of the study and I think extra work is needed to make this manuscript suitable for a publication at gchron.

Below are typographical corrections I found:

Line 138: SiO2: the "2" must be in index

Line 333: "We now can": this is actually "We can now"

Line 393: "more higher": remove "more"

Line 403: "one can": repeated twice, remove one of the two

---

## Author Comment (AC1)

**Review Sinnesael et al. Cyclostratigraphy of the Middle to Upper Ordovician successions of the Armorican Massif (western France) using portable X-ray fluorescence**

The manuscript of Sinnesael et al. present new portable XRF data obtained from an Ordovician succession from western Brittany (France). The sequence stratigraphy was previously published. The succession corresponds to the record of an open marine environment, with deposits covering from the shoreface to the lower offshore. This change in environment trigger changes in type of sediment deposited (sand vs. clay) but also change in sediment rates, with higher sediment rates in more proximal deposits.

I understand from the manuscript that the section is protected and no sampling was performed because of this reason. Measurements were thus performed in situ using spectral gamma ray and portable XRF (pXRF). Spectral analyses were performed from the potassium content approximated from the spectral gamma ray and pXRF. Two members were analyzed: the Morgat Mb., consisting of claystone deposited in a lower offshore environment, and the Kerarmor Mb., representing alternations between claystone and sandstone deposited in an upper offshore to shoreface environment.

We largely agree with the two previous paragraphs that describe some of the main aspects of the presented manuscript, although we don't fully agree with the statement in the third sentence, which does not reflect our text as such. In our interpretation changing rates of sedimentation also depend on changing sea level.

From the spectral analyses done on the K series, the authors conclude that the Morgat Mb. Contains an excellent record of the Milankovitch cycles, unlikely to the Kerarmor Mb. The authors attribute the inability of spectral analyses to identify the sedimentary record of the Milankovitch cycles in the Kerarmor Mb. to the highly unstable sedimentation rate, demonstrating that the sequences, the processes and environments at the origin of the deposits have to be identified before applying Fourier analyses, and their derivative methods, on a sedimentary series. To be franc, this conclusion brings little novelty. This is a useful reminder that cyclostratigraphy, before being a statistical challenge, is a geological challenge. If one wants to suggest to have the record of the Milankovitch cycles, they have to prove first that they have climatic and/or eustatic cycles which average period is within the range of the astronomical cycles. From a Fourier analysis, this is possible to find a frequency content that mimics the astronomical cycles, with however no climatic or eustatic significance. Relying only on a Fourier transform done from a geophysical signal is thus not enough, especially if event deposit, such as storm deposits, are included in the sampling and the frequency analysis. This shall be obvious, and one might consider this conclusion as naive. This is however sometimes forgotten in publications.

'Excellent' is maybe a too strong word for our interpretation of a potential Milankovitch signal in the Morgat Mb. Our conclusion is more carefully formulated, acknowledging that in the absence of precise independent age constraints, a Milankovitch origin is hard to demonstrate conclusively. But indeed, the more homogenous muddier facies of the Morgat Mb. did seem to show less challenges in terms of the interpretation of the spectral analysis results compared to the more sandstone-rich Kerarmor Mb..

We agree that cyclostratigraphy is in the first place a geological challenge before a statistical one, and that this is *an sich* not a new insight. Our paper explores how these theoretical insights are applied to a Paleozoic case study with its specific challenges in terms of for example non-stationarity of the signals and a lack of high-resolution independent age control. And indeed, some of the considerations described here by reviewer 1 are very often not considered or overlooked in publications, which is exactly what forms an important motivation for the presentation of this manuscript in its current form – an effort that was much appreciated by reviewer 2. We want to emphasize that the new pXRF work is also a substantial part of the manuscript and seems to have been accepted without major reservations by both reviewers 1 & 2.

As suggested by reviewer 2, a short additional paragraph discussing some examples and comparisons should strengthen this point. We suggest adding the following paragraph at the end of the current discussion (L404):

"Dealing with cyclostratigraphic uncertainties in a Paleozoic integrated stratigraphic framework is not an easy task (e.g. Sinnesael et al., 2019; Ghobadi Pour et al., 2020). Studies that, similarly, target less conventional facies in younger stratigraphical intervals might in general have more robust independent age constraints (e.g. Noorbergen et al., 2018) or more reliable astronomical parameters like insolation curves available (e.g. Vaucher et al., 2021), while this much less the case for the Paleozoic (e.g. Laskar, 2020) - often resulting in looser temporal constraints on astronomical interpretations. For example, Sinnesael et al. (2021) reinterpreted the expression of astronomically forced Upper Ordovician sedimentary cycles on Anticosti Island (Long, 2007; Elrick et al., 2013) resulting in a different interpretation of the duration of the cycles by an order of magnitude. The use of correlations and ages that only are loosely constrained, in order to imply astronomical origins of sedimentary sequences, is not uncommon when interpreting lower Paleozoic records (e.g. Sutcliffe et al., 2000; Gambacorta et al., 2018). Other common practice is the application of spectral techniques on stratigraphic records that might not be ideal for such type of analysis because of, e.g., their variable lithologies and associated variable expression of the proxies used (e.g. Zhong et al., 2018). These challenges accentuate the need for further developed cyclostratigraphic methodologies that are not simply a copy of what has been shown to work well for younger stratigraphic intervals; instead we need techniques that are adapted to the reality of the more limited availability of accurate independent age constraints and the lack of well-preserved open marine pelagic sections that characterize the Paleozoic sedimentary record."

Saying that, I find the spectral analyses of the Kerarmor Mb. superficial, in a sense that, knowing that the time of deposit of sandstone beds can be regarded as instantaneous compared to the time needed to deposit the claystone beds, the authors should try spectral analysis of the K content on the section "sandstone-free", i.e. removing data and thickness of the sandstone beds from the series to only keep the decantation deposits, which sedimentation rate and variability of K content was probably much more stable. In complement, the authors could use the study of Dabard et al. (2015) to convert the sequences they attribute to the precession cycles to they expected average period (so they make an orbital tuning from the sequencing of Dabard et al., 2015) to remove the variations in the sedimentation rate obviously depending on the sandstone beds only.

Indeed, the sedimentation rate of sandstone beds is closer to instantaneous compared to the mudstones. The removal of 'event beds' (e.g. turbidites, volcanic ashes, small slumps, …) is not

ideal, but a not uncommon practice in cyclostratigraphy. In this case, however, removing the sandstones from the sandstone-dominated Kerarmor Mb. would mean removing most of the stratigraphy… Reviewer 2 actually states *"—, avoiding some fiddling that most often are nonsense (e.g. removing sandstones from the succession, only keeping the clay… and forgetting that each base of a sandstone bed is an erosional surface remobilizing cms to tens of cms of shales)."* We briefly reiterate our conceptual understanding of the depository mechanisms for these storm-dominated deposits:

1)      HCS arenites and also intercalated argillites are first transported and then deposited by storms. The type of transport, and consequently the time of the deposition process of each layer, is a function of grain size.

2)      The stratigraphy of storm-dominated terrigenous deposits, and thus the representation of the time, consists largely of sedimentary voids. Layers are laid down when sediment is available and not all storms produce layers. Therefore, layers are produced by the successful storm that produces deposition.

3)      Clay deposits, which are deposited in the lower offshore of the same terrigenous platform described above, are also produced by storms with the same process of intermittent deposition that is a function of successful storms. These deposits also possess the same time gaps not represented by sediment , as described for the previous.

The variation of K is function of grain size which controls the mineralogy. Removing the sand removes the main indicator of bathymetry variation.

The general aim of the manuscript was to show difficulties of applying and interpreting basic spectral analyses when certain underlying conditions of these statistical techniques are not met (e.g. non-stationarity of the sin waves, for example caused by large changes in sedimentation rates). More advanced forms of spectral analysis that build upon the basic techniques might therefore also not be valid.

Although it is at first sight an attractive suggestion to use the Dabard et al. (2015) astronomical interpretations for a tuning; we fear there is the danger for circular reasoning to then use the tuned series to test for a potential Milankovitch origin of the original signal.

In general in this manuscript, I find the graphic representation of the spectra unclear. The description of the spectra is also extremely superficial.

We regret that the power spectra are perceived to be unclear, and are open for more specific suggestions on what is not clear and what can be improved. The EHA figures used in the manuscript look very similar to the evolutive Fourier transform Figure 2 presented by the reviewer.

We add the following sentences to additionally describe and clarify the spectra in more detail in words:

L295: "Here, we use evolutive harmonic analysis (EHA) to evaluate the spectrum of the signal as it evolves throughout the stratigraphy. The frequencies that explain more variation within the moving window will have higher spectral power and are shown by redder colors (Fig. 5B). This approach has the advantage over a single periodogram or multi-taper spectrum that it can also be used to evaluate how stratigraphically consistently present a certain period might be or not. Spectral analyses indicate two main periodicities: a longer one of ~1.5-2.0 m (0.5-1.0 cycles/m) and a shorter one around ~0.5 m (1.8-2.2 cycles/m) cycle thickness (as indicated by the dotted lines on Fig. 5A). The rest of the EHA shows very little elevated spectral power for other frequencies."

L303: "Compared to the NGR EHA, the pXRF EHA suggests additional frequencies with lower spectral power (Fig. 5). These seem, however, less stratigraphically continuous."

I redid the spectral analyses. Below are the spectrum and the power spectrogram of the K content in the Morgat Mb. It appears that the frequency the authors chose (1.5 m, 0.5 m and 0.3 m) are not the highest powers or confidence levels regarding a red noise. Can the authors explain their choice? Is it based on their stratigraphic continuity? What is the origin of the other frequencies?

We appreciate the effort of the reviewer to redo some of the spectral analyses. We do, however, not fully agree that there is a large difference between both analyses. The most significant peak from the reviewer's Fig. 1 is 0.25 m, which is in perfect correspondence with our statement on L303 '*and new dominant cyclicity around 0.25 m (Fig. 5F).*' – and so not 0.3 m. The 1.6 m peak is very close to our quoted ±1.5 m peak. It is correct that the ~0.5 m peak is not very present in the pXRF K spectrum, just like we mention in L302-303: '*, a less pronounced 0.5 m cycle*'. We mention the 0.5 m cycle here, because it appeared earlier on in the spectrum of the lower resolution NGR data. A lot of the 'other frequencies', are actually close to 0.5 m. Indeed, an important argument for the choice of the mentioned frequencies is their (partial) stratigraphic continuity as evaluated by an evolutionary type of analysis, where only looking at the total spectrum of a signal might have the risk of highlighting spectral peaks with high significance power, but that actually do not appear in large parts of the section. The issue of statistical testing of spectral peaks in cyclostratigraphy is also subject of debate, as for example debated in the Vaughan et al. (2011) reference suggested by the reviewer. In summary, both ways of spectral analyses seem thus consistent with each other – we also provide all the data and script we used in the supplementary materials.

[Figure]

Figure 1: 2π-MTM spectrum of the pXRF K content from the Morgat Mb. Purple line is median smoothing; brown line 90 % confidence level; green line 95 % CL and pink line 99 % CL.

[Figure]

Figure 2: Evolutive Fourier transform of the pXRF K content from the Morgat Mb. The red lines are spectral peaks. The blue color indicates the spectral background. The Fourier transform were done on 5-m intervals.

In the chapter of the spectral analysis of the Kerarmor Mb., the author apparently experienced difficulties in calculating the long-term trend, which surprised me. In short sections, it may indeed not be trivial, however in this case, applying a locally weighted scatterplot smoothing curve with a coefficient of 0.5 allows the lowest frequencies to be decreased to low values while preserving the spectral peaks at higher frequencies. Notice that with this procedure, no spurious peak is produced at low frequencies, following the recommendation from Vaughan et al. (2011).

It is correct that for the short sections of the Kerarmor Mb. and Morgat Mb. we only applied a linear detrending. The topic of the importance of different levels of detrending is actually discussed in detail for the analysis of the whole Postolonnec Fm. (e.g. Fig. 7 and L361-371) and further discussions. One of the goals of this manuscript is to highlight the challenges and pitfalls of using classical spectral analysis tools on records that do not respect the underlying assumptions needed to apply these techniques in the first place.

[Figure]

Figure 3: Detrending of the pXRF K content of the Kerarmor Mb. Top right figure is the spectrum before detrending (only the average of the series is set to 0 for clarity). Bottom right is the spectrum after detrending applying a LOWESS with a coefficient of 0.5.

In summary, I find the description and the design of the experiment extremely superficial, and additional work is needed in my opinion. So, at the moment I am not convinced by the design of the study and I think extra work is needed to make this manuscript suitable for a publication at gchron.

We regret this opinion. The first main concern raised by reviewer 1 is that there is no real contribution in discussing challenges and pitfalls of using classical spectral analysis tools on records that do not respect the underlying assumptions – while this is a real problem in the current literature, as appreciated by reviewer 2. To bring this point forward more clearly, we add a short paragraph discussing some relevant examples and the end of the current discussion. We believe that our detailed reply regarding suggested differences in results between our spectral analysis and some presented by reviewer 2 demonstrates that the results are in essence comparable. We also add some additional description of the spectra in the main text to make them clearer.

Below are typographical corrections I found:

Line 138: SiO2: the "2" must be in index   Implemented

Line 333: "We now can": this is actually "We can now"   Implemented

Line 393: "more higher": remove "more"   The meaning should have been 'additional' instead of 'more'. This was indeed unclear and has now been changed.

Line 403: "one can": repeated twice, remove one of the two   Implemented

---

## Author Comment (AC2)

The manuscript *gchron2021-45* by M. Sinnesael is a (very) well written, well documented and well-presented case study describing efforts in order to decipher astrocycles from a Lower Paleozoic succession, for which a strong sedimentological background is available from a series a previously published papers. Corresponding to an epicontinental archive (no oceanic record for the considered time interval!), relatively shallow marine deposits are here considered with different, yet combined signatures (depositional facies, pXRF, gamma-ray); the results are of course somewhat disappointing, as the case is as expected challenging. It is good —and rare— to read a paper where the authors remain scientifically 'honest', not hiding but highlighting the many problems —even if, in theory, they are well known by the community—, avoiding some fiddling that most often are nonsense (e.g. removing sandstones from the succession, only keeping the clay… and forgetting that each base of a sandstone bed is an erosional surface remobilizing cms to tens of cms of shales).

According to me, the paper could be accepted with only a minor revision required. Below some typos and suggestions for further improving the manuscript and its impact.

We thank reviewer 2 for appreciating one of the major messages of the paper, and acknowledging that discussing the presented issues – even though they should be known problems – is very valuable for the community.

Below we reply to each minor suggestion individually:

Line 42: pre-Cretaceous (don't forget the Precambrian record)   Implemented

Line 68: Geol. setting and BIOstratigraphy   Implemented

Line 70: a comma is needed after Fig. 1A   Implemented

Line 71: on? the cliff   Change 'on' into 'at'.

Line 86: a few words about the total related time duration according the chronostratigraphic time scale would here have been welcome (in addition to lines 255-260), to have in mind the temporal significance of the studied interval.

We added the following sentence to better situate the general reader:

"Overall, the Postolonnec Fm. spans roughly fourteen million years starting close to the start of the Darriwilian Stage (~467 Ma) and ending close to the end of Sandbian Stage (~453 Ma)."

Line 185: it means that some of the 'stratigraphic surfaces' with nodules are hiatial surfaces with zero 'averaged' accumulation, indicating also high variations even in shales successions. Need a short discussion?

We agree that there is *stricto sensu* no such thing as 'continuous sedimentation', with also in shale successions possible variations in sedimentation, including no deposition, therefore we would suggest adding: ", indicating that even in the more mudstone-dominated intervals changes in

sedimentation rates can be pronounced." Also see our reply to Reviewer 1 on suggestion to remove sandstone intervals.

Line 193… that can fossilize evoluting sedim. facies (?) Changed to 'preserve'.

Line 198 also mention the usual absence of event beds in the more distal successions

Added: "Usually, more distal successions feature also much less event beds."

Line 231: I would have expected a few word about the mineralogy of the placer deposits, either on the basis of thin sections, or XRF analyses: relationships discussed here need to refer to relative contents in zircon, rutile, or monazite (or others)

We now 1) refer to previous work focusing on the mineralogy of these placers "e.g. zircons, monazite and titaniferous minerals as described by Pistis et al., 2016; 2018" and 2) added following sentence (L222): "Our 'single spot per bed' measurements do not allow for specific mineralogical identifications, but an approach that combines multiple pXRF measurements for the same sample for coarse-grained igneous rocks has shown promising results to extract mineralogical information (Triantafyllou et al., 2021)."

Line 250: … is, at this stage, only suggestive…

We believe that the current formulations are already careful and specific enough in their current form.

I would have like to see clear location of placer deposits (e.g., arrows or stars positioned adjacent to the logs)

Excellent suggestion, we have now added these to Fig. 4.

[Figure]

Figure 4: 'Previous caption' + The black stars indicate the stratigraphic occurrence of paleoplacers.

Lines 284-287: if the first element (=i) clearly relates to methodological considerations, the second (=ii) is more a cautionary note or at least a welcome reminder.

We partially reformulated towards "we consider two additional elements to the cyclostratigraphic analysis of the Postolonnec Fm.:".

Line 299: … to have have even more (e.g., Martinez et al., 2016)   Implemented

Line 318-319: Even though… to be rephrased?

Indeed, unclear formulation, rephrased now "It is hard to further demonstrate an astronomical origin of these variations in the absence of more precise stratigraphic constraints."

Line 320-320: this short comment would have merited some more explanations and/or hypotheses. It may suggest that even in offshore shale-dominated successions, remobilization may occur, removing/displacing nodules (then behaving as clats); or that condensation may not be systematically expressed by nodules.

Assuming reviewer 2 refers to Lines 321-322; We tend to think more of this as the latter suggestion. Nodules are interpreted to be an expression of condensation, but there might be various boundary conditions at play, so that there is indeed not always a one-to-one relationship. The conditions of early diagenesis vary with the sedimentation rate, which moves the sediment-water interface more or less rapidly, and all this strongly conditions the stratigraphic result of the nodule concretion. References Loi and Dabard (2002) and Dabard and Loi (2012) studied these aspects in more detail. We suggest adding the following sentence: "That nodules occur less systematically might be related to the interplay of various boundary conditions (as discussed in Loi and Dabard (2002) and Dabard and Loi (2012)) that can lead to a varied sedimentological expression of a condensation interval."

Line 335: wouldn't be better to write something as (?): "the sequences of similar dominant frequencies are…"

We prefer to keep the current formulation as L334-336 is more descriptive, while indeed in our next sentence (L336-337) we mention that these might be related to the same sequences, but here there is already a more interpretative component.

Lines 355-356: would it need a verb?

We added 'they might be'.

Line 364: The expected changes in sedimentation rates are not counted, in this context, in a few percent, but more likely spread over 2 or 3 orders of magnitude (give references?): this should be better underlined here.

Good comment. This is also extremely hard to quantify, especially here where age constraints are virtually absent. Therefore, we would suggest adding 'rates (up to an order of magnitude)' L364.

Lines 380-383: a little too much optimism? Such Zr grains are rare, with a time interval of several million years between crystallization and deposition...

We suggested to add the following complementary discussion: "A conceptually related approach concerns the dating of prismatic zircons in a Darriwilian limestone bed in Sweden (Lindskog et al., 2017; Liao et al., 2020). For both the detrital zircon and non-bentonite associated prismatic zircon dating approaches it is crucial to keep in mind that the numerical age from the dating comes with an additional (larger) uncertainty on its depositionary age. Even when such uncertainties would be in the order of millions of years they can still be valuable in such cases where there are tens of millions of years worth of little chronometrically constrained stratigraphy."

One should also not undervalue a 'maximal depositional age'. Besides its dating uncertainty, the specific dated horizon can really not be older. When looking at zircon ages from, for example, a bentonite, one should indeed also consider aspects such as lead loss (often well dealt with nowadays) or if the zircon crystallization ages do correspond with the actual eruption age. When using biostratigraphy one also has to consider that first or last occurrences of a species/assemblage may not be well identified, may not be true FADS or LADS, may not really be globally synchronous etc…

Lines 395-400: more frequent occurrences of event beds in proximal, sand-dominated interval, which are not or poorly tied to astrochronological controls, might here be emphasized

Thank you for this insight. We suggest adding the following sentence to L393: "In general, there is also a higher occurrence of event beds in more proximal and sandstone-dominated intervals, which are most probably not astronomically influenced."

The conclusion is relatively 'flat'. The utility and superiority of pXRF appeared clearly demonstrated. It would have been also welcome a statement about potential durations of the Dabard et al.'s (2015) cycles: are they only confirmed? Better characterized? With new proposed durations/controls? Is a revision necessary?

We suggest adding the following more explicit sentences to the conclusion (in agreement with the ending of the abstract): "In the absence of precise independent age constraints, it is not possible to fully confirm the astrochronological framework suggested in Dabard et al. (2015). A potential astronomical signal is suggested in the more homogenous mudstone facies, while reliable cycle identification in the more proximal sandstone-dominated intervals proofs to be challenging."

Finally, a short paragraph including a comparison with conclusions of other published Lower Paleozoic case studies (e.g., Long, 2007, Can. J. Earth Sci. **44**: 413–431 or Elrick et al., 2014: *Geology* 2013;41;775-778 among other) or, alternatively, with more recent successions displaying records featured by similar lithologies (e.g., Vaucher et al. doi.org/10.1038/s41598-021-96372-x ) would have strengthen the paper.

We will follow this suggestion, also as it relates to some of the comments raised by reviewer 1. We suggest adding the following paragraph at the end of the current discussion (L404):

"Dealing with cyclostratigraphic uncertainties in a Paleozoic integrated stratigraphic framework is not an easy task (e.g. Sinnesael et al., 2019; Ghobadi Pour et al., 2020). Studies that, similarly, target less conventional facies in younger stratigraphical intervals might in general have more robust independent age constraints (e.g. Noorbergen et al., 2018) or more reliable astronomical parameters like insolation curves available (e.g. Vaucher et al., 2021), while this much less the case for the Paleozoic (e.g. Laskar, 2020) - often resulting in looser temporal constraints on astronomical interpretations. For example, Sinnesael et al. (2021) reinterpreted the expression of astronomically forced Upper Ordovician sedimentary cycles on Anticosti Island (Long, 2007; Elrick et al., 2013) resulting in a different interpretation of the duration of the cycles by an order of magnitude. The use of correlations and ages that only are loosely constrained, in order to imply astronomical origins of sedimentary sequences, is not uncommon when interpreting lower

Paleozoic records (e.g. Sutcliffe et al., 2000; Gambacorta et al., 2018). Other common practice is the application of spectral techniques on stratigraphic records that might not be ideal for such type of analysis because of, e.g., their variable lithologies and associated variable expression of the proxies used (e.g. Zhong et al., 2018). These challenges accentuate the need for further developed cyclostratigraphic methodologies that are not simply a copy of what has been shown to work well for younger stratigraphic intervals; instead we need techniques that are adapted to the reality of the more limited availability of accurate independent age constraints and the lack of well-preserved open marine pelagic sections that characterize the Paleozoic sedimentary record."

We also add "which is sometimes also applied in similar studies (e.g. Loi et al., 2010)." to L401.